# Sensor Information Sharing Using a Producer-Consumer Algorithm on Small Vehicles

**DOI:** 10.3390/s21093022

**Published:** 2021-04-25

**Authors:** Rodrigo Vazquez-Lopez, Juan Carlos Herrera-Lozada, Jacobo Sandoval-Gutierrez, Philipp von Bülow, Daniel Librado Martinez-Vazquez

**Affiliations:** 1Instituto Politécnico Nacional, Centro de Innovación y Desarrollo Tecnológico en Cómputo (CIDETEC), Unidad Profesional Adolfo López Mateos, Ciudad de México 07700, Mexico; rvazquezl1800@alumno.ipn.mx (R.V.-L.); jlozada@ipn.mx (J.C.H.-L.); 2Departamento de Procesos Productivos, Universidad Autónoma Metropolitana Unidad Lerma, Estado de México 52005, Mexico; p.von@correo.ler.uam.mx (P.v.B.); d.martinez@correo.ler.uam.mx (D.L.M.-V.)

**Keywords:** absolute position system, cooperative algorithm, intercepting vehicles, indoor positioning, robot framework, UWB sensors

## Abstract

There are several tools, frameworks, and algorithms to solve information sharing from multiple tasks and robots. Some applications such as ROS, Kafka, and MAVLink cover most problems when using operating systems. However, they cannot be used for particular problems that demand optimization of resources. Therefore, the objective was to design a solution to fit the resources of small vehicles. The methodology consisted of defining the group of vehicles with low performance or are not compatible with high-level known applications; design a reduced, modular, and compatible architecture; design a producer-consumer algorithm that adjusts to the simultaneous localization and communication of multiple vehicles with UWB sensors; validate the operation with an interception task. The results showed the feasibility of performing architecture for embedded systems compatible with other applications managing information through the proposed algorithm allowed to complete the interception task between two vehicles. Another result was to determine the system’s efficiency by scaling the memory size and comparing its performance. The work’s contributions show the areas of opportunity to develop architectures focusing on the optimization of robot resources and complement existing ones.

## 1. Introduction

The use of an operating system has made it possible to standardize the technologies for the problem of cooperativity between several robots or vehicles. To this end, the devices, the sensors, the communication modules, the protocols, and the programming languages are designed to be compatible with each other. This conception of unified platforms such as ROS, Kafka, AEROSTACK, MAVLink among others, resolves several hardware problems and allows a more user-friendly environment for software developers [1,2,3,4]. However, high compatibility leads to a significant increase in resource usage. Another consequence is to lose the diversity of solutions that are the result of particular case investigations.

In this research, a case study shows the importance of maintaining research with custom designs than generic designs. A basic platform scenario requires resources at three fundamental layers. The perception layer (indoor systems) uses a GoPro-type internal camera [5,6] or a VICON type external camera sophisticated system [7,8]. The communication layer involves the use of transmitters, routers, and standardized protocols such as WiFi, Zigbee, or Bluetooth [9]. The third layer of applications involves the use of a compatible information management system, i.e., master and slave workstations using a computer architecture with an operating system such as Ubuntu or similar. The above implies a high energy consumption in managing information compared to the energy required for actuators who achieve vehicles’ displacement. However, it is essential to consider the advantages of these solutions. For example, end-users, developers, and interested people can program solutions in a more friendly way.

In another perspective, there are tailor-made solutions so specific, complex, and customized that they require specialized equipment and are not compatible with other systems. Therefore, the proposal is based on a methodology that combines customized solutions with generic ones to be implemented in low-performance vehicles or incompatible with operating systems. For this, a reduction or replacement of tools is proposed without losing compatibility with high-level technologies. Given the characteristics of this architecture, algorithms based on hardware or technology constraints should be implemented. Finally, the actual functionality should be validated through a known task and where the optimization of resources can be compared.

A significant difference between the design of a generic and a customized platform lies in taking advantage of the sensor’s characteristics. In particular, UWB sensors allow sending messages and obtaining the position by triangulation. Consequently, known tools cannot be applied directly, and an intermediate algorithm has to be used. The libraries of programming languages based on operating systems are very advanced compared to embedded systems. It is important to remember that the migration of a tool or algorithm of custom technology to a general one is always possible. However, in the opposite direction, it is not always guaranteed. Therefore, it is essential not to miss the opportunity to investigate code optimization for embedded systems.

### 1.1. Contribution

The design and implementation of an alternative architecture for small vehicles is the main contribution of this work. Simultaneous localization and sharing information with a UWB technology reduced computing resources and consumption energy. A suitable producer–consumer algorithm gives a solution to manage the concurrent processes in reduced-scale hardware. Other contributions are listed as follow:Enable incompatible hardware with high-level architectures.Manage information packets between embedded systems.Achieve an interception task among vehicles avoiding the use of additional hardware or complex infrastructure.Make compatible an architecture that uses operating systems with one that does not.Adapt to the confinement conditions when performing experiments at home.The small vehicle platform allows test different autonomous navigation strategies, lowering the risk and cost of large-scale testing.

### 1.2. Organization

The article is organized as follows. Section 2 presents a brief review of the state-of-the-art in robotic systems architectures. The customized architecture for small vehicles and its underlying concepts are presented in Section 3. The experiments, results and discussions are developed during Section 4. Conclusions and future Work can finally be found in Section 5.

## 2. Background and Related Work

### 2.1. Robotic System Architectures

Software-oriented solutions in multi-vehicle tasks have consistently been standardized on architecture with at least three layers with particular purposes. These layers’ functions are the quantification of physical variables, the communication between sensors and processing devices, and the applications that solve the tasks [10].

This architecture seeks to centralize the sensors’ information in an integrated manner on the devices running the applications. Sophisticated communication systems are used to guarantee the exchange of information. The processing and storage devices use operating systems to program the solutions to the tasks. The above clearly shows that the software-oriented perspective allows for more excellent compatibility. However, poor performance or incompatibility can occur with specific problems.

#### 2.1.1. Localization Sensors

The Inertial Navigation Systems (INS) are fed with data from their Inertial Measurement Unit (IMU), composed mainly of sensors such as gyroscopes and accelerometers. INS are cost-effective solutions that do not require additional infrastructure. The main disadvantage of these systems lies in the cumulative position error that grows during the operation of the vehicle [11]. However, the combination of the information from an INS with a global positioning system allows to improve the accuracy of the positioning estimates [12].

A variety of technologies applied to indoor global positioning systems have been sought. The results in [13] showed that accuracy depends on the implementation and methodology used. In particular, the technologies, which were applied in the development of autonomous vehicles, are global positioning systems by vision such as VICON [7,8], or Optitrack [14,15], Virtual Reality (VR) systems [16,17] and RF technology [18,19,20]. Meanwhile, the use of Ultra-Wide Band (UWB), part of the RF technologies, attracted interest during the last years [21], because with this technology it is also possible to obtain a good performance in terms of accuracy [22].

Typical tasks that autonomous vehicles must fulfill depend on the quality of the information about the vehicle’s absolute and relative position within its workspaces, such as Simultaneous Localization and Mapping (SLAM), obstacle avoidance, trajectory tracking, and cooperation. The integration of absolute and relative positioning systems is, therefore, necessary. The most recent research efforts present various technologies to achieve a good precision in detecting the global and relative position and thus perform a satisfactory SLAM. On the other hand, UWB and vision systems have been combined to improve the estimation of the position in the workspace [23]. Other strategies bet on the fusion of UWB data with those coming from INS [24], and Optitrack [25]. Additionally, it has been tried to improve UWB systems’ results on a platform of mobile robots using Gaussian processes [26] and neural networks [27].

#### 2.1.2. Communication Protocols

Indoor Wireless Communications are technologies that allow devices to be uniquely identified and exchanged over a limited distance of 10 m to 100 m [9]. There are several criteria for selecting Bluetooth, UWB, ZigBee, or Wi-Fi technology. The most commonly used technologies are Bluetooth or WI-FI because most robots have it by default. ZigBee is identified as an alternative for users looking to work with a more significant number of devices, a low transfer rate, and requiring less energy consumption. Finally, UWB technologies are less present as a combination of high transfer rate and low normalized energy consumption.

#### 2.1.3. Applications

Applications such as ROS, Kafka, and MAVLink are versatile tools to incorporate most devices, infrastructures, programming languages, and technologies. For example, ROS is defined as a robotic middleware to help manage the complexity and heterogeneity inherent in distributed systems [1]. Apache Kafka is a distributed messaging system widely used in big data applications [2] and MAVLink is a communication protocol used for the bidirectional communications between drones and ground stations over a wireless channel [4]. In this sense, by having approved research platforms, it is possible to leave aside hardware problems and focus on algorithms’ performance.

There are different examples of applications that use ROS in its architecture as in [28,29,30]. In [31] Apache Kafka was used to parallelize a set learning architecture. In contrast, in [29], small drones are used to collect data to test obstacle avoidance algorithms. Finally, MAVLink being a communication protocol, can be found implemented in several tools such as ROS. In [32] MAVLink was used with ROS to communicate with the pixhawk drone and perform autonomous flights. In [33], they used the protocol to communicate with real drones to validate the performance of the proposed architecture.

The element relationships of the Robotic System Architectures with the software developed are shown in Table 1.

Table 1 shows that the proposed architecture incorporates non-compliant devices such as EV3 devices and UWB technology as localization, communication, and infrastructure devices.

Some architectures propose using cloud services to improve the computational capacity and intelligence in vehicles such as AGVs [35]. Some examples suggest using architectures based on the edge-fog model as seen in [30]. The authors used the publish-subscribe model of ROS to perform 3D scenario reconstruction through SLAM and proof of concept presented in [36]. In the case AVG [37] the authors give an application-oriented to AGV collaboration, while in [38] they use ROS. Other similar examples can be seen in [39,40,41].

#### 2.1.4. Cooperative Tasks

Cooperative tasks have been investigated in various scenarios with software-oriented architectures [42,43]. Information sharing using algorithms among aerial or ground vehicles has been studied. These algorithms are proposed to solve the path planning problem [44], for the surveillance of multiple moving ground targets [45] or intercepting intruders [46]. Tools of general-purpose have proposed the simulation of these algorithms. However, real implementation of the hardware of these algorithms requires a particular adaptation. The adaptation to solve this problem is shown in Section 3.

## 3. Customized Architecture for Small Vehicles

The previous section showed a typical software-oriented architecture with the most common technologies. In this section, an alternative hardware-oriented architecture will be developed. Figure 1 shown a general comparison of layers and technologies employed.

The small vehicle platform allows test different autonomous navigation strategies, lowering the risk and cost of large-scale testing. The platform enables destructive testing at a meager cost, infrastructure and energy consumption reduced.

### 3.1. Small Vehicles Models

#### 3.1.1. Ground Robot

We used a robot as a mobile reference point within the workspace. The robot with 3-DOF (x,y,θ) is of a differential configuration, so it has two independent, active wheels and a fixed wheel as a support point. The odometry of the vehicle uses encoders and a gyroscope as a relative reference system. The vehicle’s operation is executed intrinsically in the controller using the dynamic model proposed in [47].

Figure 2 shows the robot with its differential system variables as it travels along a given path in an absolute coordinate system $X_{A}Y_{A}$. The mobile is composed of two wheels with diameter ϕ, each placed at a distance L from the intermediate point P. The velocities of the wheels are denoted by $v_{d}$ (dextram = right) and $v_{s}$ (sinestram = left) respectively, so that the system inputs are:(1)$$\left[L,\phi ,\theta ,v_{s},v_{d}\right]\in R^{5}.$$

According to the model of direct kinematics presented in [47], the robot motion is given by the following matrix equation in (Equation 2).
(2)$$\xi_{A}=\left[\begin{array}{c} (\omega_{d}+\omega_{s})cos\theta \\ (\omega_{d}+\omega_{s})sin\theta \\ \omega_{d}-\omega_{s} \end{array}\right],$$

#### 3.1.2. Aerial Vehicle

A 6-DOF nano-quadcopter (*x*,*y*,*z*, pitch, roll, yaw) to follow the ground reference robot’s trajectory was used. The internal odometry consisted of INS, whose IMU integrates an accelerometer, gyroscope (in three axes) and a barometric pressure sensor. Additionally, the INS data are merged with the height estimation obtained employing an optical flow sensor and the absolute positioning system’s data. The operation of the vehicle is done in an autonomous, semi-autonomous way or by using teleoperation.

To operate the vehicle, we assumed the 12-state model, studied in [48,49] and validated in [50]. According to Figure 3, the model requires two reference systems: the non-inertial one $X^{\prime }Y^{\prime }Z^{\prime }$ located at the center of gravity of the quadcopter, orientated as shown, and the inertial one *x*, *y*, and *z* relative to the center of the earth. Of the 12 parameters, the variables *x*, *y* and *z* represent the coordinates of the center of gravity, *u*, *v* and *w* are the linear velocities along the $X^{\prime }$, $Y^{\prime }$ and $Z^{\prime }$ axes, and ψ, θ and ϕ represent the rotation angles (pitch, roll and yaw) with the corresponding angular velocities. All variables are expressed within the inertial reference system. The model is non-linear and is described in [50,51].

### 3.2. Sensor Information Sharing

A particular research objective was proposed a 3D global positioning platform oriented to devices with limited hardware. The low-cost UWB sensors were used to perform simultaneous localization and information exchange between the small vehicles.

The system consists of anchors and tags that communicate with UWB pulses. The anchors within the workspace serve as absolute reference points for the tags. Likewise, the tags are used as a mobile reference point to triangulate the anchors’ signals and, simultaneously, as a communication interface to other external devices. Two-Way Ranging (TWR) algorithms are used to detect a single tag, and a Time Difference of Arrival (TDoA) algorithm is used to detect the position of one or more tags inside the same workspace. The diagram of the operation is shown in Figure 4.

### 3.3. Producer-Consumer Algorithm

A reduced algorithm based on an analogy to the producer–consumer problem was another particular research objective. This consideration helps to perform tasks to share information with two or more complex sensors to localize small vehicles. Furthermore, this algorithm is reduced to achieve accessible communication among multiple robots and an extra path planning task was executed at the same time.

Collaboration in navigation among autonomous robots represents a problem of synchronization between the transmission and reception of positioning and movement data. In this problem, there appear two parallel processes (producer and consumer), which exchange information by a finite memory buffer [52]. Here the producer is in charge of generating and inserting data into the buffer, trying not to saturate it. Otherwise, the consumer extracts the data from the buffer one by one, preventing the buffer from emptying. Applications of this problem are commonly in concurrency, and message passing, although trivial, is extremely useful when needed [53]. Figure 5 shows the flowchart of the consumer–producer problem.

The producer–consumer problem is somehow analogous to the problem of cooperative follow-up of a trajectory. In said problem, one or more dependent robots follow the trajectory generated by a master robot until finally coinciding in (almost) the same point. The master robot draws the trajectory in the workspace, so it operates as a producer of information on points. On the other hand, the dependent robots receive their information on the trajectory needed to get to the master, beginning from their starting points. It is, though, possible to view the dependent robots as consumers of information.

The FIFO (First In-First Out) buffer can be modelled as an ordered set *D* of $q_{i}$ information elements. The state of this kind of memory is determined through the sequence $\left\{q_{0},q_{1},\cdots ,q_{m}\right\}$, ordered by the moment insertion of its elements by (Equation 3) and (Equation 4). In this buffer, a new information element $q_{0}$ is always placed at the end of its queue by the insertion instruction of an $I(q_{0},D)$:(3)$$I\left(q_{0},D\right)=q_{0}\cup D.$$

The operation raises all elements’ position in the buffer by one place and increases, consequently, the cardinality by one unit: $I(q_{0},D)|=|D|+1$. On the other hand, the instruction of an extraction procedure E(D), removes, when executed, the first element $q_{m}$ from the queue of the m data elements in *D*:(4)$$E\left(D\right)=D\backslash q_{m}.$$

It is clear that this command lowers the cardinality of the set *D* by one unit: |E(D)|=|D|−1.

For implementation of the algorithm in the robots, we used the following general considerations. Let $W\subset R^{3}$ be the workspace and R the set of positions of the *i* robots, with $i\in Z^{+},\;i=1,\ldots n$ and n>1. The value of the instantaneous position at time *t* of each robot is represented by $R_{i}\left(t\right)=\left(x_{i}\left(t\right),y_{i}\left(t\right),z_{i}\left(t\right)\right)$. The trajectory ${}_{t_{0}}^{t_{f}}R_{i}$ of the robot *i* between the starting point $R_{i}\left(t_{0}\right)$ and the endpoint $R_{i}\left(t_{f}\right)$ are characterized by a sequence of positions $R_{i}\left(t_{j}\right)$ which corresponds to the equidistant sequence of sampling moments $t_{j}=t_{0}+j\cdot \Delta t$, within the interval between the initial $t_{0}$ and final time $t_{f}$, separated by a sampling interval Δt between each moment, such that: ${}_{t_{0}}^{t_{f}}R_{i}=\left\{R_{i}\left(t\right)\right\}\forall t\in \left[t_{0},\ldots ,t_{f}\right]$. To the master robot we assign, as identifier, the number 1. In this way the path of this robot is symbolized with ${}_{t_{0}}^{t_{f}}R_{i}$. The elements of the trajectory are then successively in $D_{1}$ loaded elements with the insertion command:(5)$$I(R_{1}\left(t_{j}\right),D_{0}).$$

The follower robots $R_{s}$, with s∈{2,3,..,n}, reconstruct the path by successively extracting the leading elements in the $D_{1}$ buffer:(6)$$R_{s}\leftarrow \left(D_{1}\right).$$

The Algorithm 1 for the case of the master robot (producer) can be summarized as follows:
**Algorithm 1** Producer algorithm1:**procedure**Producer($D_{1},trajectory$)2:  Initialize $|D_{1}|=0$ (empty)3:  $R_{M}$ starts trajectory at $R_{1}\left(t_{0}\right)$4:  **while**
trajectory is not finish **do**5:    $R_{M}$ inserts its position in memory $D_{1}$6:    Delay Δt

The Algorithm 2 for the case of the follower robot (consumer) can be summarized as follows:
**Algorithm 2** Consumer algorithm1:**procedure**Consumer($R_{M},D_{1}$)2:  **if**
$R_{M}$ starts trajectory at $R_{1}\left(t_{0}\right)$
**then**3:    **while** position $R_{1}\left(t_{j}\right)$ is changing or $D_{1}$ is not empty **do**4:     Extract an element from $D_{1}$ and send it to robot s, as its next position $p_{s}$5:      Move to $p_{s}\left(x_{s},y_{S},z_{s}\right)$

## 4. Experiments, Results and Discussion

### 4.1. Experiments

The experiments performed were divided into two parts: (1) the characterization of the sensor information sharing with small vehicles and (2) intercepting and landing of the aerial vehicle on the ground vehicle with shared information using the producer–consumer algorithm.

#### 4.1.1. Experiment 1: Characterization

This experiment aimed to characterize the absolute positioning system and, after that, estimate the tracking accuracy in 2D and 3D. The system was placed within a controlled environment, and functionality tests were carried out within the fixed position’s polygon, defined by the anchors. The 2D tracking tests consisted of following prior defined trajectories using the ground vehicle. The corresponding robot was tagged to know in real-time its position in the xy plane, left side of Figure 6. In the xyz space, we used a nano quadcopter with a tag integrated, as shown on the right side of Figure 6. The data obtained was used to reconstruct the trajectory and to know the accuracy of the system.

We used a Loco Positioning System (LPS), which employes a UWB DMW1000 module from Decaway and whose precision is estimated by the manufacturer to be within ±0.1 m. The LPS is composed of a Loco Positioning Node (anchors) and a Loco Positioning Deck (tags). For triangulation of the signal, the system supports configurations with 4, 6 and 8 anchors and uses the TWR algorithm (a single tag) and TDoA in the existence of two or more tags. A minimum of four anchors must triangulate the signal within a 3D space, so we selected this configuration during our experiments. Four anchors (numbered from 0 to 3) were placed at a distance 0.16 m above the floor, forming a rectangle with dimensions *m* by *n*. We decided to place the absolute reference system’s origin in the center of this m×n rectangle. Table 2 shows the values of *m*, *n* and the coordinates of each anchor.

A LEGO EV3 ground robot was used in its differential configuration. We employed a gyroscope with an accuracy of ±3 degrees and an optical encoder with an accuracy of ±1° to measure the wheels’ rotation. The controller of the EV3 ground robot contains an ARM 9 processor with 64 MB of RAM and an SD card reader on which the GNU/Linux-based EV3dev Operating System (OS) was installed. The trajectories and odometry of the EV3 land mobile as described in [54] were implemented using Python as the programming language and the EV3dev OS libraries.

The airbone vehicle used is a Crazyflie 2.1 quadcopter, which has as main controller an STM32F405 and uses the nRF51822 for communication with the Crazyradio-PA telemetry system. The architecture of the Crazyflie quadcopter allows for the stacking of accessories, known as decks. We used the Flowdeck V2 as an optical flow sensor and the LPS Deck as a tag, compatible with LPS. The Crazyflie quadcopter’s architecture is open to implement tools and algorithms made by the community of developers. Crazyflie quadcopter uses the Kalman Extended Filter, as proposed by [55,56] for the fusion of data of the IMU, the Flowdeck V2 and the LPS. The implemented control algorithms are based on [57,58].

For the individual tests, we worked with the TWR mode. We assigned tag 1 to the Crazyflie quadcopter and tag 2 to the LEGO EV3 robot. As a test to determine the UWB triangulation system’s accuracy, we implemented a desired trajectory in the workspace for both robots. A computer was interfaced with a Crazyradio-PA to acquire the position data of each tag. For this purpose, a circle of 0.61 m radius and different heights, 0.02 m in the EV3 robot and 0.5 m in the Crazyflie quadcopter.

#### 4.1.2. Experiment 2: Intercepting and Landing Task

The experiment’s objective was to validate the operation of the proposed consumer-producer algorithm when solving the combined task of intercepting and landing, where the Crazyflie 2.1 quadcopter must intercept the LEGO EV3 lander and land on top of it. The initial position of the EV3 robot was *x* = 0 m, *y* = 0.66 m and *z* = 0.02 m, and its given path was an arc with a radius of 0.5 m and an arc length of also 0.5 m. The initial position of the Crazyflie quadcopter was at *x* = 0 m, *y* = 0 m and *z* = 0 m, and during the trajectory tracking phase, the quadcopter was asked to maintain a height of 0.5 m.

Figure 7 shows the flowchart of both the producer and consumer algorithms implemented on a computer to manage the dataflow of the EV3 robot, the Crazyflie quadcopter and the TDoA mode of the LPS. In this case, process A (producer) is the EV3 robot, and process B (consumer) is the Crazyflie quadcopter. The process starts when the computer has verified the communication between the EV3 robot and the Crazyflie quadcopter. The computer sends then to the EV3 robot the information about the trajectory to be followed. When the EV3 robot starts to move, it sends its position data to the computer. In parallel to the EV3 robot following the demanded path, the computer starts the producer-consumer algorithm and instructs the Crazyflie quadcopter to start its takeoff. When the Crazyflie quadcopter reaches the reference height of 0.50 m, it starts receiving the buffer position data of the EV3 robot from the computer.

The EV3 robot sends its position data continuously and independently to the $D_{1}$ buffer in the computer until it reaches the endpoint of the given trajectory. The Crazyflie quadcopter consumes the position data added by the EV3 robot in the buffer $D_{1}$ and computes its movements by comparing its position with the buffer’s positions. At the moment, the Crazyflie quadcopter does not encounter any more position data in the buffer $D_{1}$ it starts a landing maneuver and touches down on top of the EV3 robot.

The details about the algorithm’s implementation can be viewed in the block diagram in Figure 8. It illustrates the relationship of the tools used between the computer, the EV3 robot and the Crazyflie quadcopter during the algorithm’s execution. The white arrows indicate the internal communication flow, while the dotted arrows represent external communication flow.

The algorithm was implemented on a Python script using multi-threaded programming and executed on a computer with a Core i7 processor with 8 GB of RAM. The Python script sends orders over the local network to the EV3 robot using the MQTT protocol through an MQTT broker installed in the EV3dev OS in the EV3 robot. MQTT was used because it is ideal for machine-to-machine (M2M) communication due to this protocol’s lightness. On the other hand, the EV3 robot was connected with a wireless network card.

The communication between the UWB tags and the computer was done using the Crazyradio-PA telemetry system, which communicates wirelessly, using the NRF24 protocol. One Crazyradio antenna can communicate with up to 8 tags.

#### 4.1.3. Experiment 3: Performance

This experiment aimed to evaluate the scaling of the system about the size of the consumer’s buffer. The proposal consisted of generating a constant amount of information of 1.92 kbps by the producer. The tested trajectory was the same as in the previous experiments with a speed of 0.1 m/s. Buffer size started with 0, increments of 1 and up to 14 times (24 Bytes up to 360 Bytes). The units of measurement used were: the consumption rate in Kbps, the kinetic energy in $\mathrm{Kg}\cdot m^{2}/s^{2}$, and the processing time of the information in the queue with s. The break-even point to determine the system’s efficiency was to compare the increase of the energy and the processing time.

An additional test was to eliminate the information sent by the Ev3 robot to destabilize the operation of the quadcopter to 50% of the task.

### 4.2. Results and Discussion

During the characterization process for the anchor positions, as presented in Table 2 of Section 3, it was observed that a square-shaped distribution of the anchors in the workspace leads to a better performance than a rectangular one (Figure 9). Nevertheless, the manufacturer recommends a rectangular layout. The best configuration was a square workspace of 2 m by 2 m and with the anchors at its vertices after trials.

Figure 10 shows the results obtained after characterization and tuning of the 2 m × 2 m workspace. Figure 10 shows the ideal trajectory (dashed line) and the one performed (green line) by the EV3 robot (green line). We found that the data obtained were consistent; however, when the robot moved near the anchors, the error increased significantly up to 0.1 m. Figure 10b,c show the EV3 robot’s displacement on each of the *x* and *y* coordinate axes. The path carried out by the EV3 robot in the *x*-axis is much closer to the given one than in the *y*-axis, due to the tag’s orientation. However, there are probabilities of perturbations that increase the error during some moments, for example, in *t* = 19.7 s.

Figure 10 shows the RMS-error calculation of the path in both coordinate axes. In the *x*-axis, the error is within a range below 0 to 0.1 m as claimed by the manufacturer of the UWB system; however, in the *y*-axis, the maximum error went up to 0.16 m. The average RMS-error in the *x*-axis was 0.042 m, again by a factor of 2/3 lower than the RMS-error of 0.062 m in the case of the *y*-axis.

The producer-consumer algorithm always kept the equilibrium of data. For this reason, the Crazyflie quadcopter intercepted the EV3 robot with acceptable performance. Figure 11 illustrates the best result we obtained when executing the algorithm; that is, it represents the sequence of the points $R_{1}\left(t_{j}\right)$ that were stored in $D_{1}$’s queue, coming from the EV3 robot, which was later on converted into set points for the Crazyflie quadcopter. Figure 11a shows the trajectories performed by the EV3 robot (red line) and by the Crazyflie quadcopter (blue line) in 3D perspective. The ideal trajectory for the quadcopter is shown by the black dashed line. The projections of the trajectories on the *x*, *y*, and *z* axes are similarly represented in Figure 11b–d.

Figure 12 shown the RMS-positioning-error of the EV3 robot and the Crazyflie quadcopter from the ideal path. We restricted the evaluation of the EV3 robot’s performance to the xy-plane. In Figure 12a, it is seen that the RMS-error in *x* is always smaller than 0.22 m. When $(t_{f}-t_{0})<10$ s the error is greater than 0.1 m, for 10 s $<(t_{f}-t_{0})<$ 19 s you get an average RMS-error of 0.04 m, and in the interval of 19 s $<(t_{f}-t_{0})<$ 23 s the RMS-error is greater than 0.04 m but always less than 0.1 m, due to the proximity to the anchor three.

Figure 12b shown that the RMS-positioning-error in *y* is greater than the one in *x*. This error is because the distance between the two robots at the beginning is 0.61 m and is interpreted as an error. The Crazyflie quadcopter manages to reduce its RMS-positioning-error to less than 0.1 m after the first 6 seconds. Once a level of proximity under 0.1 m is reached, the average RMS-error in *y* drops down to a level of 0.032 m.

Figure 13a shows the performance results of the system about the scalability of the consumer buffer. The exponential behavior of the data rate with three segments can be observed. A moderate slope was observed in the first segment from 0 to 8 times the buffer size, from 8 to 12 a steeper slope and after 12 a sudden increase.

The results have allowed us to verify that the system can be scaled in terms of buffer size to increase the consumption data rate. The limitation of this increase is affected by the response of the quadcopter. Another scenario would be to increase the number of quadcopters sharing the information; in this sense, the rate is divided among the robots cooperating in the same task. The maximum operating limit of the platform is 6.8 Mbps and a packet length of 1023 bytes.

The results to find the balance point between the energy consumed by the quadcopter and the execution time are shown in Figure 13b. It was observed that with higher efficiency of the consumption data rate by the quadcopter, the performance and energy increases; however, for a rate lower than three and higher than 8, the results showed an excess or lack of time.

From the results, we can deduce that there is a relationship between the energy consumption of the quadcopter and the information efficiency. For this, by repeating the experiments, a buffer with the size of 6 is the optimal value for executing the task.

In buffer size 12, a perturbation was tested by eliminating information from the producer. Thus, it was verified that the lack of information decreased the energy consumption; therefore, the relationship between the quality of the information and the energy consumption was verified.

Figure 14 shows the results of the trajectories based on the scalability of the buffer size. A more stable behavior was observed when the information to be consumed is higher Figure 14(0–8). However, when the data is missing, there is a more unstable behavior as is shown in Figure 14(12).

Finally, a video of the experiments can be seen at https://rodrigovazquezlopez.github.io/Cooperation/ (accessed on 23 April 2021).

## 5. Conclusions and Future Work

The research highlighted the advantages of software-oriented robotic system architectures; however, the disadvantages with small vehicle resources were shown. The proposed hardware-oriented design succeeded in solving a small vehicle system’s particular problem without losing the advantage of compatibility with widely used tools.

The producer-consumer algorithm successfully solved the problem of synchronization of messages shared by two or more vehicles. This algorithm was implemented in real hardware, while other similar algorithms only were simulated. This research also solved the problem of intercepting an airborne vehicle’s trajectory to a level below 0.04 m and landing it safely on top of the ground vehicle. The EV3 robot and the Crazyflie quadcopter are affordable small-scale platforms and offer the experimenter reduced scale equipment that avoids unnecessary exposure of people to risks.

It was possible to reduce the number of anchors from 6 and 8, as mentioned in similar publications, to only 4. The square-shaped workspace gives better results than the recommended rectangular one. Furthermore, in parallel, perpendicular or diagonal paths, the RMS-positioning-error is reduced, compared to the one in curved paths, i.e., when the tags rotate in the plane, has to be interference that increases the error. The positioning error increased due to the proximity between a tag and anyone of the anchors. Even under these higher noise conditions, the results were satisfactory; that is, an average error of 0.04 m was achieved.

The LPS platform configuration has the following advantages: Experimental test in small environments of only 2 m × 2 m × 1 m (width, length and height), the platform is portable and does not require complex installations. Due to the ongoing epidemic, it is impossible to work in research laboratories in many places in the world. Therefore, the possibility of having a compact platform, like the one presented here, allows continuing developing research in our home-laboratories.

Performance tests showed the ability to scale the memory size and thus process more information. However, the efficiency of the system does not improve proportionally. A balance point was found to define a trade-off between data, power consumption, and processing time. Therefore, we can suggest considering the efficiency relationship between the size of the vehicles and their information management systems with the system’s scalability.

For further development, a new intelligent and robust algorithm will be proposed to be executed intrinsically in the airborne and terrestrial robot and achieve the algorithm’s implementation with a more significant number of cooperating robots.

## Figures and Tables

**Figure 1 sensors-21-03022-f001:**
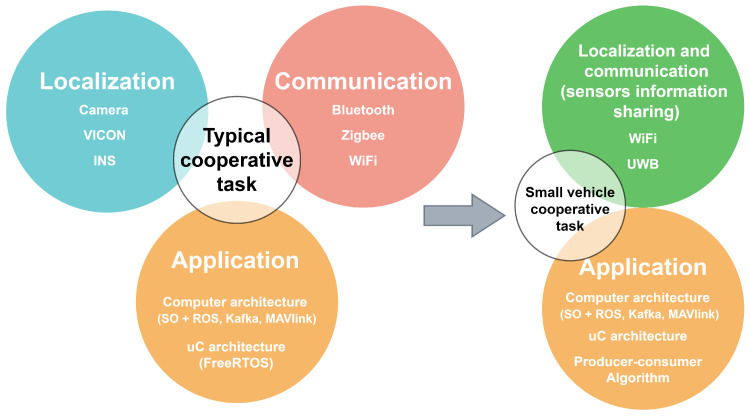
Comparison of software-oriented architecture (**left-side**) and proposed architecture (**right-side**).

**Figure 2 sensors-21-03022-f002:**
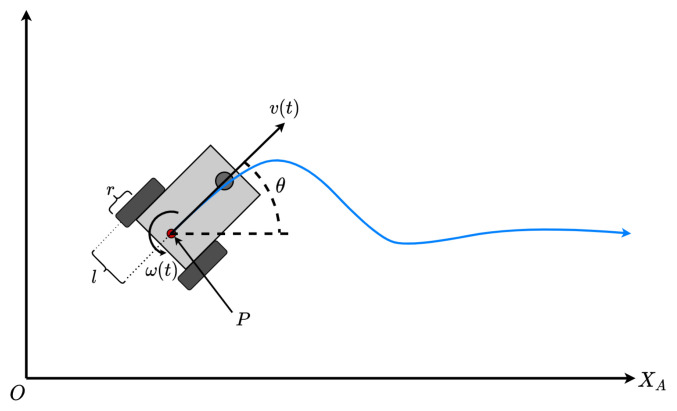
Ground vehicle model.

**Figure 3 sensors-21-03022-f003:**
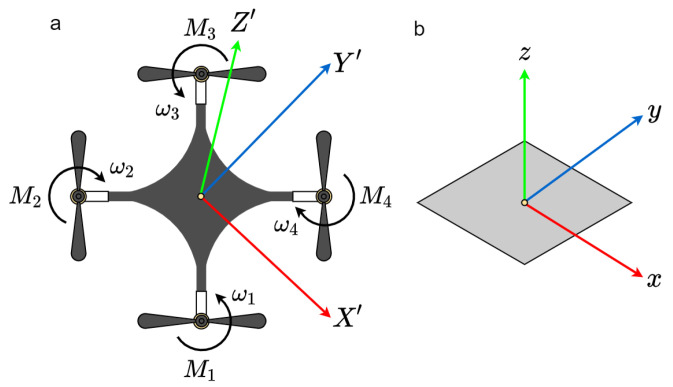
Nano-quadcopter. (**a**) Body frame. (**b**) Inertial frame.

**Figure 4 sensors-21-03022-f004:**
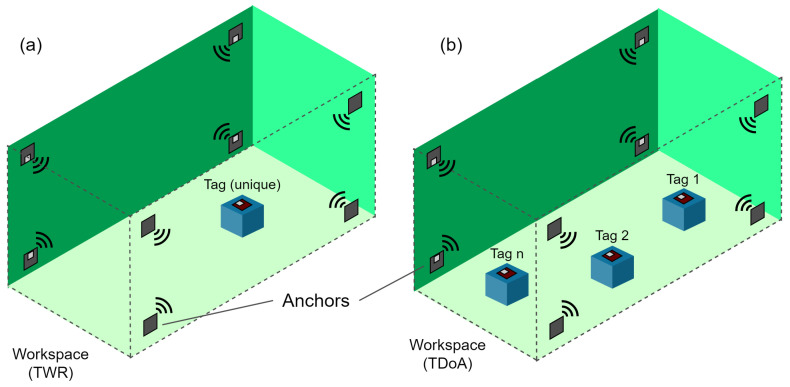
Different configurations of the absolute positioning system mounted within a workspace: (**a**) existence of only one tag: a location algorithm based on “two way ranging” TWR is used. (**b**)Two or more tags are present: a location algorithm based on “time difference of arrival” TDoA is used.

**Figure 5 sensors-21-03022-f005:**
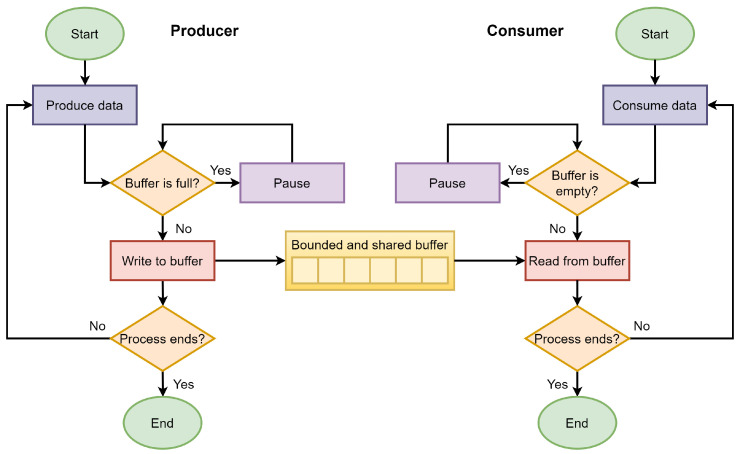
Flowchart of the producer–consumer problem.

**Figure 6 sensors-21-03022-f006:**
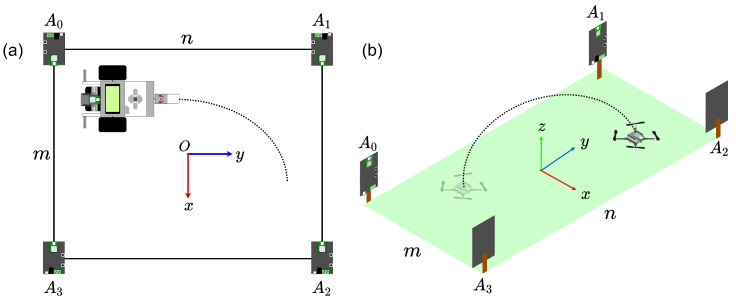
Diagram of the characterization experiment. (**a**) In 2D. (**b**) In 3D.

**Figure 7 sensors-21-03022-f007:**
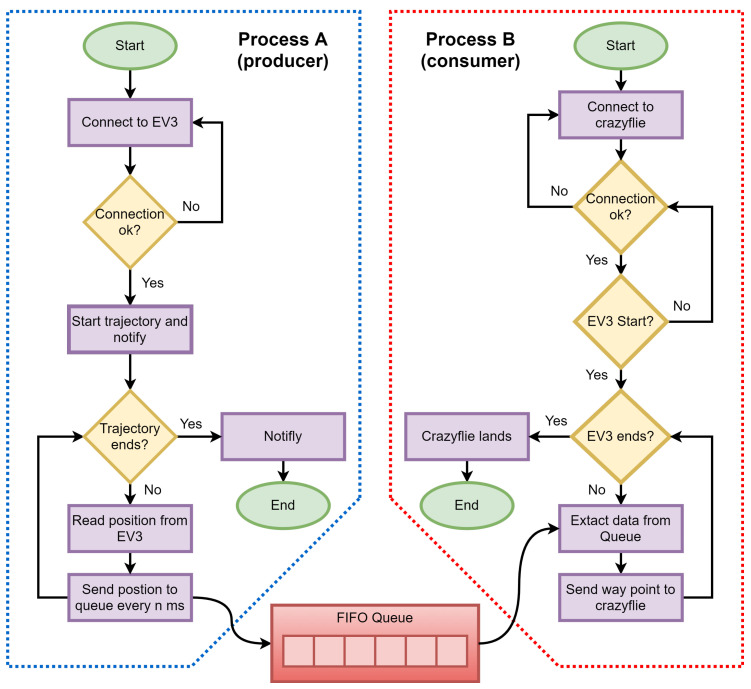
Flowchart of the process A (producer) implemented for the mobile robot EV3 and of the process B (consumer) implemented for the Crazyflie quadcopter.

**Figure 8 sensors-21-03022-f008:**
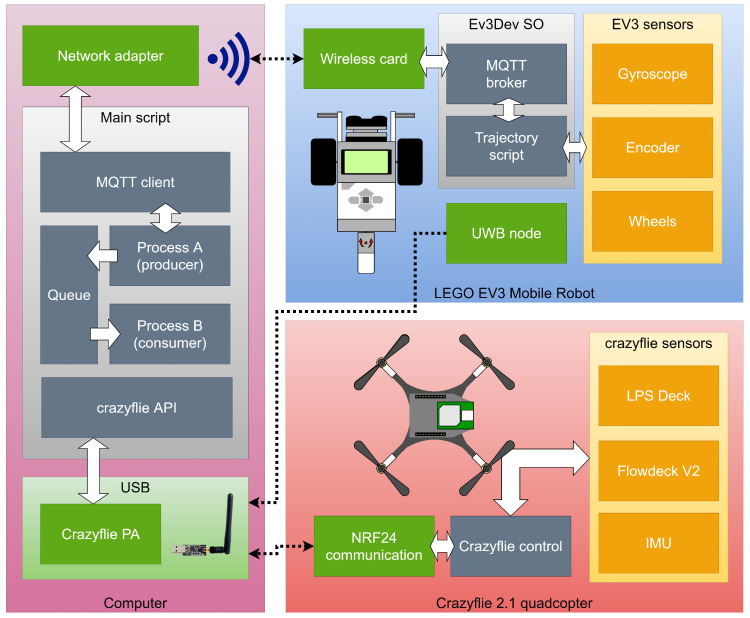
Block diagram illustrating the interaction between the components during the execution of the algorithm.

**Figure 9 sensors-21-03022-f009:**
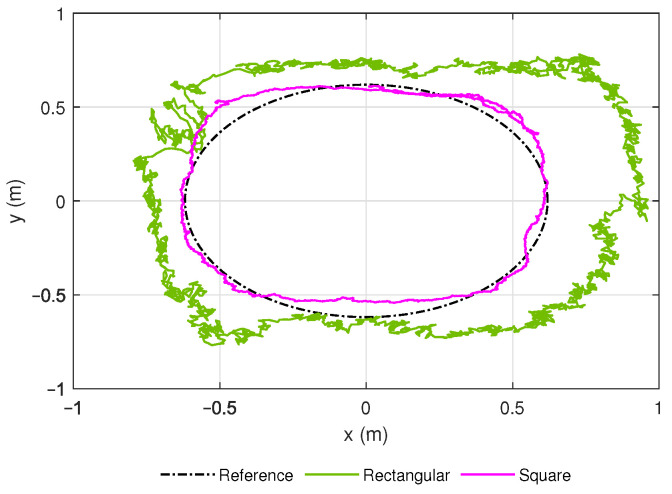
Performance of anchors in rectangular (green) and square (pink) configuration concerning a reference path.

**Figure 10 sensors-21-03022-f010:**
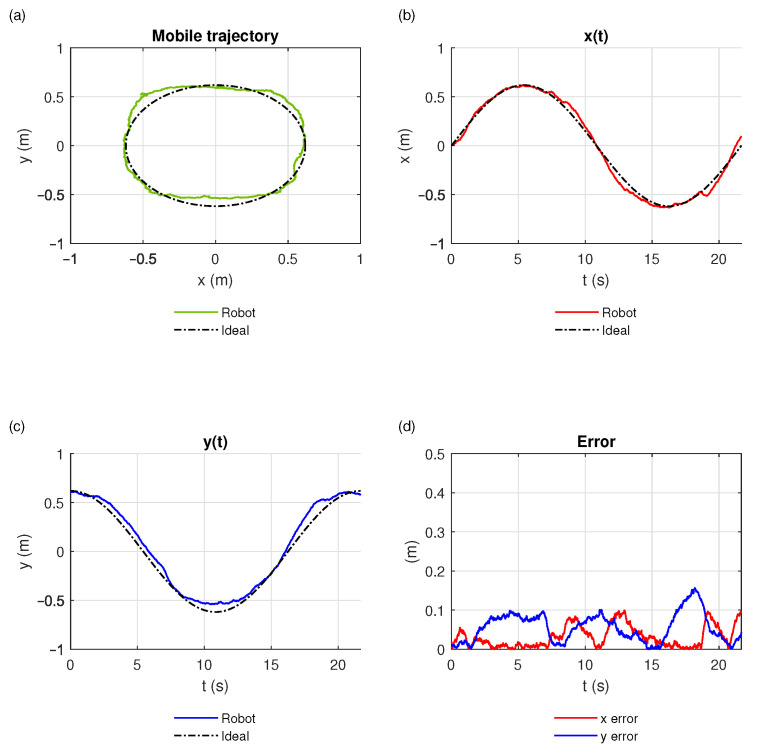
Results of the characterization run for a 2 × 2 m workspace. (**a**) Trajectory performed by the robot. (**b**) *x*-axis component of the movement. (**c**) *y*-axis component of the movement. (**d**) Quadratic error on *x* and *y* between the performed path and the given ideal trajectory.

**Figure 11 sensors-21-03022-f011:**
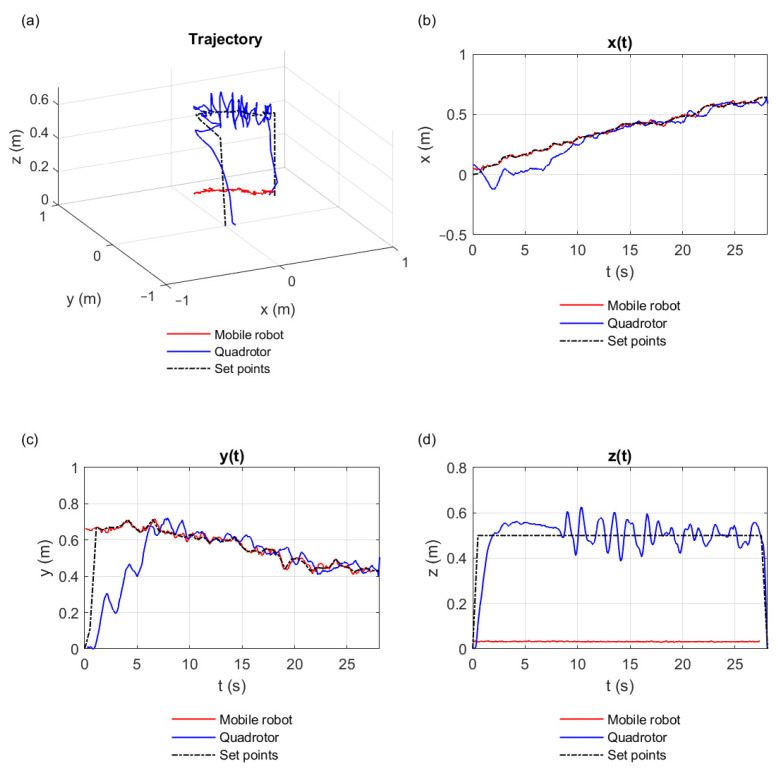
The paths taken by the EV3 (blue line), and the Crazyflie quadcopter robot (red line) compared to the ideal path (black dashed line). (**a**) The trajectory performed in 3D perspective. (**b**) Displacement in *x*. (**c**) Displacement *y*. (**d**) Displacement in *z*.

**Figure 12 sensors-21-03022-f012:**
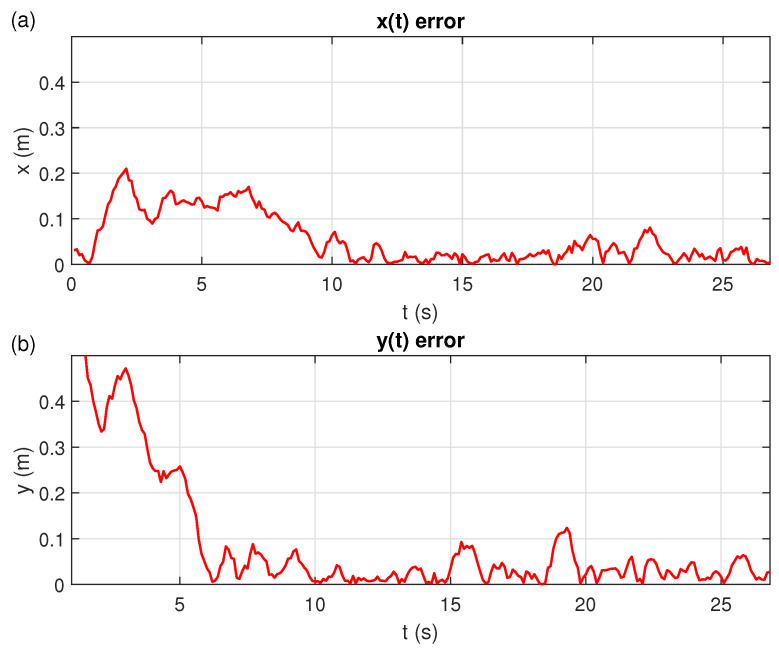
RMS-positioning-error between the EV3 robot and the Crazyflie quadcopter. (**a**) *x*-axis error. (**b**) *y*-axis error.

**Figure 13 sensors-21-03022-f013:**
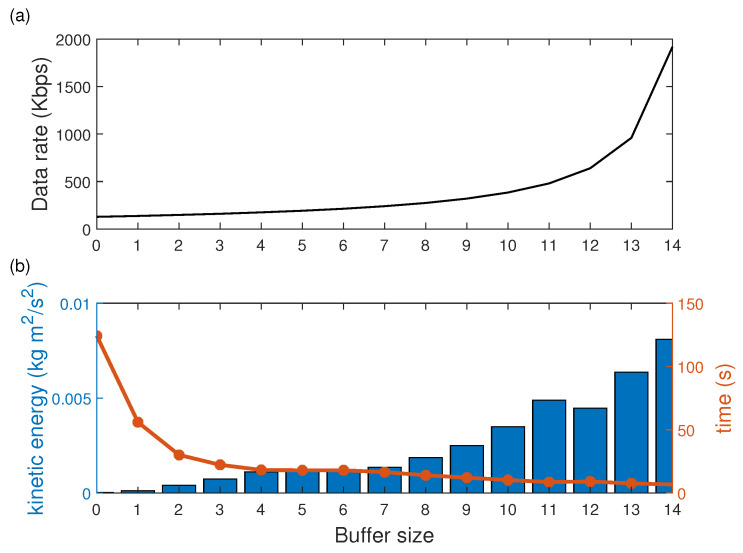
Buffer size vs. (**a**) consumption data rate and (**b**) kinetic energy and execution time comparison as a system’s performance evaluation.

**Figure 14 sensors-21-03022-f014:**
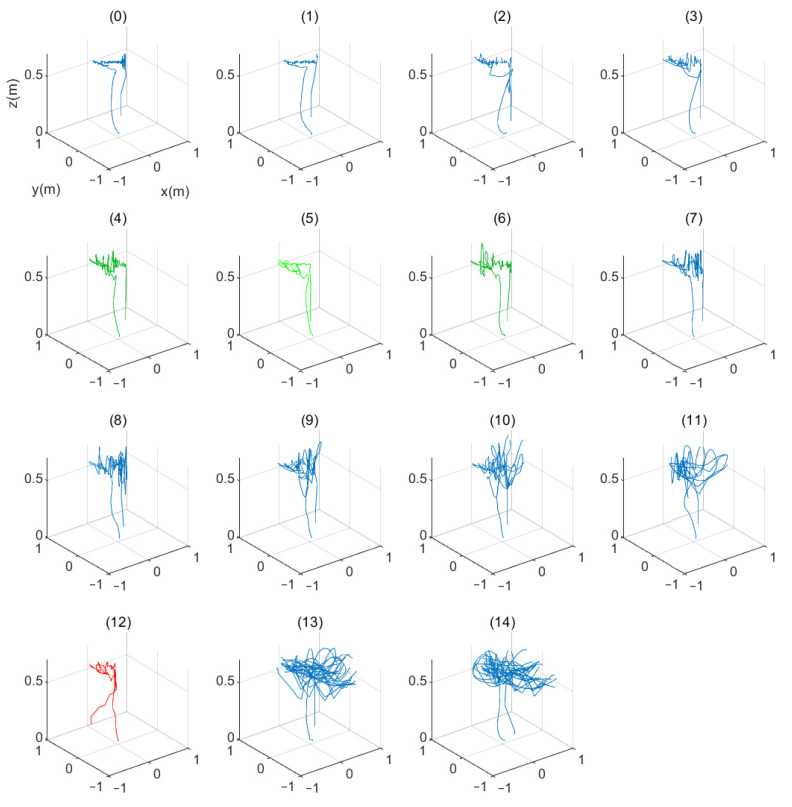
Different tests made using buffer sizes from 0 to 14.

**Table 1 sensors-21-03022-t001:** Comparison between similar applications and tools.

Software (Tool)	Computer Architecture	Supported System	Comm. Devices	Infrastructure
ROS [28]	x64 ARM	GNU/Linux	Wi-Fi	Network Camera FCU
Kafka, ROS [29]	ARM	GNU/Linux	4G	Network VPN Cloud Service Cameras
MAVLink [34]	ARM	Android 4.3	Zigbee	X-bee Modules Tablet Camera
Proposed architecture	x64… STM, EV3	Linux… μC Based	Wi-Fi	Network UWB

**Table 2 sensors-21-03022-t002:** Positions of the anchors employed to characterize the positioning system.

Size		Anchor’s Coordinates (x,y,z)			
**m [m]**	**n [m]**	**Anchor 0 [m]**	**Anchor 1 [m]**	**Anchor 2 [m]**	**Anchor 3 [m]**
1.86	5.0	(−0.93, −2.50, 0)	(0.93, 2.50, 0)	(−0.93, 2.50, 0)	(0.93, −2.50, 0)
1.90	5.0	(−0.95, −2.50, 0)	(0.95, 2.50, 0)	(−0.95, 2.50, 0)	(0.95, −2.50, 0)
5.20	1.9	(−2.50, −0.95, 0)	(2.50, 0.95, 0)	(−2.50, 0.95, 0)	(2.50, −0.95, 0)
2.00	2.0	(−1.00, −1.00, 0)	(1.00, 1.00, 0)	(−1.00, 1.00, 0)	(1.00, −1.00, 0)

## Data Availability

Not applicable.

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
