# Peer review of "Sensor Information Sharing Using a Producer-Consumer Algorithm on Small Vehicles"

_sensors, 2021, doi:10.3390/s21093022_

Round 1
Reviewer 1 Report
The authors propose a producer-consumer algorithm to share information in two positioning systems. The paper is well presented. However, I would like to know more scientific contributions in this paper. Hence, I would suggest the following improvements. 1. The authors should elaborate the scientific contributions in this paper, especially in the part of introduction. 2. How will the algorithm work on more than two positioning systems? 3. How does the inertial sensors and UWB fused in the positioning system?Author Response
Thanks for your comments. We attached a point-by-point response.
*The changes are highlighted in yellow or text in green color.

Reviewer 2 Report
The article presents a producer-consumer based algorithm for the positioning of small vehicles.
The main motivation of this study is to carry out different autonomous navigation strategies for vehicles, reducing risks and costs derived from larger tests.
General comments:
In general, the research is not well justified from the aforementioned motivation. The development and experiments carried out raise the doubt of being scalable to conditions of autonomous navigation of real vehicles, lacking a real sense of application.
The workspace used in the experiments and the configuration of sensors seems difficult to extrapolate to larger-scale experiments.
The precision of the results is not negligible considering the general dimensions of the test/robots/workspace as well as the short duration of the same.
Specific comments:
- Line 34 and 35: "Autonomous vehicles use Inertial Navigation Systems (INS) to accomplish the task of relative positioning." It is not very precise and ambiguous, since they not only use this technique for relative positioning.
- Line 48: References 18 and 19 have little relevance for the statement where they are cited.
- Section 2: The Producer-Consumer algorithm does not bring much innovation
Author Response
Thanks for your comments. We attached a point-by-point response.
*The changes are highlighted in yellow or text in green color.

Author Response
|
Comments and Suggestions for Authors |
Answer |
Pages |
|
The paper addresses a topic of importance for local authhorities and population. Usage of ANFIS models is acceptable tool. Nevertheless the authors must provide more technical information, related to exactly the inputs used in their study and the difference between the fuzzy and the neural network model they used. |
This paper is entitled “Sensors Information Sharing using a Producer-Consumer Algorithm on Small Vehicles” |
No changes |
Reviewer 4 Report
The authors have proposed to create a suitable algorithm capable of operating in real indoor conditions with multiple information sharing. As an example, they have used two small vehicles operating in different domains (air+ground).
Although the paper was somewhat written with suitable text and figures, its proposal, design, and results are not well coupled.
What was proposed was an algorithm to share multiple information while the results focus on the application without entering in details about the methodology.
Therefore, important information and tests were not performed;
- what are the differences (and advantages) of your proposal agains ROS or KAFKA?
- How can you ensure reliablity and security?
- What are the scalabity?
- Is the system depends on the architecture? (server/cliente, edge only, fog only, edge-fog)
- For simple client/server peer-to-peer systems there are the mavlink and protobuf approaches that works well.
- there are several multi-robots, multiple task frameworks that provides better and more in-depth analysis (aerostack, proactive, arcog, ffiduas, etc.)
Therefore, it is necessary to focus on the methodology once the application is not new, and there is no novelty on what was presented as application.
Author Response
Thanks for your comments. We attached a point-by-point response.

Reviewer 5 Report
This paper focus on an important issue with analogy based method. Extensive experimental results with hardware implementation have been provided to validate the efficiency of proposed method.
However, this paper was prepared more like a technique report than a scientific research paper.
Section 2 discuss too much on a well studied algorithm, which contribute with almost no information. It will be more important to discuss why you utilize producer-consumer algorithm, and how it was embedded into your algorithm.
Similarly, section3 discusses the system design rather than the information sharing. Although the system design is important and should be discussed as well, it can not instead the algorithm discussion.
The results provided in the experiment section should be classified as application level results, e.g. location accuracy, rather than the information sharing metrics. The direct observation on the information sharing efficiency should be provided.
I will suggest an re-orgnization of this paper focus on the information sharing algorithm, e.g. what’s the problem of state of the art solution? Why you choose this algorithm? Any modification has been made? Why you implement such modification to suits such scenarios?
Author Response

(The authors gave the same response as above.)

Round 2
Reviewer 2 Report
Improvement in the quality of the paper is appreciated.
However, there are still several points that need to be better detailed. Mainly there are two aspects to improve: the section where experiments are described and clarify the scalability of the proposal.
Author Response
|
Comments and Suggestions for Authors |
Answer |
Pages/Lines |
|
Improvement in the quality of the paper is appreciated. However, there are still several points that need to be better detailed. Mainly there are two aspects to improve: the section where experiments are described and clarify the scalability of the proposal. |
A new subsection with an scalability experiment was made. This modification is visible in Sections: Abstract. 4.1.3 Experiment 3 4.2 Results and discussion 5. Conclusions |
(Blue and Highlighted in yellow) 1 / 11-13
12 / 352-361 16/ 403-424 17 / 425-428 17/ Figure 13 18 / Figure 14 19 / 457-462 |

Reviewer 4 Report
The authors have improve the manuscript, however some points must be better analyzed;
- mavlink does not need an operational system as stated at the answer.
- ROS and KAFKA can be used as an external server to orchastrate messanges among the agents in a edge-fog or fog-cloud architecture.
- scalabity is a key factor for your proposal and it must be considered. There are several papers that deals with this distribuited architectures for low processing robots such uavs, agvs, rovs, etc...
Author Response
|
|
||
|
Comments and Suggestions for Authors |
Answer |
Pages/Lines |
|
The authors have improve the manuscript, however some points must be better analyzed; |
|
|
|
mavlink does not need an operational system as stated at the answer.
ROS and KAFKA can be used as an external server to orchastrate messanges among the agents in a edge-fog or fog-cloud architecture.
There are several papers that deals with this distribuited architectures for low processing robots such uavs, agvs, rovs, etc…
|
This point was studied in depth to understand it with new references. Sentences were added |
(Highlighted in yellow)
3-4 / 135-142 4 / 147-153 |
|
scalabity is a key factor for your proposal and it must be considered. |
A new subsection with an scalability experiment was made.
This modification is visible in Sections:
Abstract. 4.1.3 Experiment 3 4.2 Results and discussion 5. Conclusions
|
(Blue and Highlighted in yellow)
1 / 11-13
12 / 352-361
16/ 403-424 17 / 425-428 17/ Figure 13 18 / Figure 14
19 / 457-462 |

Reviewer 5 Report
I appreciate the authors work on the efforts of modification.
However, the experiment section may still need improvement. As I was mentioned, the current results were not direct observations on the information sharing efficiency or accuracy. The authors should provide direct measurements on the information sharing not the location accuracy in the application level.
Author Response
|
|
||
|
Comments and Suggestions for Authors |
Answer |
Pages/Lines |
|
I appreciate the authors work on the efforts of modification.
However, the experiment section may still need improvement. As I was mentioned, the current results were not direct observations on the information sharing efficiency or accuracy. The authors should provide direct measurements on the information sharing not the location accuracy in the application level. |
A new subsection with an efficiency and information sharing experiment was made.
This modification is visible in Sections:
Abstract. 4.1.3 Experiment 3 4.2 Results and discussion 5. Conclusions
|
(Blue and Highlighted in yellow)
1 / 11-13
12 / 352-361
16/ 403-424 17 / 425-428 17/ Figure 13 18 / Figure 14
19 / 457-462 |

Round 3
Reviewer 4 Report
in my opnion, the paper may be published.